# Therapy Prospects for Mitochondrial DNA Maintenance Disorders

**DOI:** 10.3390/ijms22126447

**Published:** 2021-06-16

**Authors:** Javier Ramón, Ferran Vila-Julià, David Molina-Granada, Miguel Molina-Berenguer, Maria Jesús Melià, Elena García-Arumí, Javier Torres-Torronteras, Yolanda Cámara, Ramon Martí

**Affiliations:** 1Research Group on Neuromuscular and Mitochondrial Diseases, Vall d’Hebron Research Institute, Universitat Autònoma de Barcelona, 08035 Barcelona, Spain; javier.ramon@vhir.org (J.R.); ferran.vila@vhir.org (F.V.-J.); david.molina@vhir.org (D.M.-G.); miguel.molina@vhir.org (M.M.-B.); maria.jesus.melia@vhir.org (M.J.M.); elena.garcia@vhir.org (E.G.-A.); javier.torres@vhir.org (J.T.-T.); yolanda.camara@vhir.org (Y.C.); 2Biomedical Network Research Centre on Rare Diseases (CIBERER), Instituto de Salud Carlos III, 28029 Madrid, Spain

**Keywords:** mitochondria, mtDNA, replication, depletion, multiple deletions, therapy, nucleoside, gene therapy

## Abstract

Mitochondrial DNA depletion and multiple deletions syndromes (MDDS) constitute a group of mitochondrial diseases defined by dysfunctional mitochondrial DNA (mtDNA) replication and maintenance. As is the case for many other mitochondrial diseases, the options for the treatment of these disorders are rather limited today. Some aggressive treatments such as liver transplantation or allogeneic stem cell transplantation are among the few available options for patients with some forms of MDDS. However, in recent years, significant advances in our knowledge of the biochemical pathomechanisms accounting for dysfunctional mtDNA replication have been achieved, which has opened new prospects for the treatment of these often fatal diseases. Current strategies under investigation to treat MDDS range from small molecule substrate enhancement approaches to more complex treatments, such as lentiviral or adenoassociated vector-mediated gene therapy. Some of these experimental therapies have already reached the clinical phase with very promising results, however, they are hampered by the fact that these are all rare disorders and so the patient recruitment potential for clinical trials is very limited.

## 1. Mitochondria and Mitochondrial Diseases

Mitochondria are organelles located in the cytoplasm of most eukaryotic cells and are involved in many important cellular metabolic processes, among which the synthesis of ATP has been classically considered their main function [1]. Mitochondria also play other important cellular roles such in the maintenance of calcium homeostasis, regulation of apoptosis, lipid metabolism and the synthesis of hormones, vitamins and the heme group [2,3,4,5].

The mitochondrial pathway leading to ATP synthesis is known as oxidative phosphorylation (OXPHOS), and it is mediated by the transformation of the energy contained in the reduced substrates NADH and FADH_2_ into electrochemical energy which is finally used to phosphorylate ADP to ATP, the main energy currency in the cell. Electrons from these reduced compounds flow through different substrates by the catalysis of four complexes (I–IV) of the electron transport chain (ETC). This flow is coupled to the pumping of protons from the mitochondrial matrix to the intermembrane space, creating a proton electrochemical gradient across the mitochondrial inner membrane that finally drives the phosphorylation of ADP to ATP through the catalysis of the ATP synthase, the fifth complex of the OXPHOS system [1].

The endosymbiont hypothesis postulates that mitochondria and plastids originated from prokaryotic progenitors associated with ancestor hosts to generate the first eukaryotic cells [6], and this theory is widely accepted [7]. Owing to their origin, mitochondria have their own genome consisting of a variable copy number of a circular double-stranded molecule of DNA. In humans, mitochondrial DNA (mtDNA) is 16.5 kb and contains 27 genes encoding 13 proteins of the OXPHOS system, as well as 22 tRNAs and 2 rRNAs used for the translation of these proteins, which takes place within the mitochondria [1]. Every cell contains several mitochondria and every mitochondrion in turn contains several copies of mtDNA. The amount of mitochondria and the mtDNA copy number depends on the tissue and its specific energy demands [8].

As indicated above, 13 subunits of the mitochondrial OXPHOS system (belonging to complexes I, III, IV and V) are encoded by mtDNA. All other subunits of the five OXPHOS complexes, as well as all other proteins located within mitochondria and needed for their function (more than 1100 proteins) are encoded by the nuclear DNA (nDNA), synthesized in the cytosol and then imported to mitochondria [9]. Therefore, the operations of mitochondria are controlled by two genomes: nDNA and mtDNA, a fact that determines many aspects of the mitochondrial function and the characteristics of different mitochondrial disorders.

Although mitochondria have several functions, the term “mitochondrial disease” classically refers to those diseases caused by dysfunctional OXPHOS which results in insufficient ATP synthesis in affected tissues [5]. This encompasses a group of diseases with heterogeneous clinical presentation and genetic origin caused by mutations in either mtDNA or nDNA. The most affected tissues are the most energy demanding ones: muscle, central nervous system, endocrine system, liver and kidney [10]. The inheritance pattern of mitochondrial diseases can be autosomal dominant or recessive, X-linked (for those caused by mutations in nuclear genes) or maternal (for those caused by mutations in mtDNA) [5]. When the mtDNA mutation is located in one of the 13 genes encoding subunits of the OXPHOS system, the consequence is usually an isolated deficiency of the affected complex (in some cases, the molecular defect in one complex alters the correct assembly of other ETC components, thus affecting more than one complex). By contrast, when the mutation affects genes encoding one of the 22 tRNAs or the two rRNAs, the translation of the 13 mtDNA encoded subunits is disturbed, leading to a multicomplex defect. In addition, the inheritance and phenotype of the diseases caused by primary mtDNA mutations is determined by the polyplasmic nature of mtDNA [5].

Alternatively, OXPHOS dysfunction can be caused by mutations in nuclear genes encoding subunits of OXPHOS complexes, as well as other factors involved in several processes needed for correct mitochondrial function: factors needed for the ETC complexes assembly, factors regulating mitochondrial protein import, enzymes of the coenzyme Q synthesis pathway, factors involved in mitochondrial translation, maintenance of the mitochondrial lipid milieu or in mitochondrial dynamics, and all factors needed for the correct mtDNA replication and maintenance. Ultimately, mutations in these genes also lead to OXPHOS impairment and therefore to an insufficiency of energy production [11].

## 2. Mitochondrial DNA Depletion and Multiple Deletions Syndromes (MDDS)

Mitochondria contain multiple copies of mtDNA whose integrity and maintenance are critical for the faithful expression of all mtDNA-encoded OXPHOS subunits and correct mitochondrial function. As indicated above, some mitochondrial diseases caused by mutations in nuclear genes affect proteins needed for mtDNA replication and maintenance. When the proteins encoded by these genes do not work properly, dysfunctional mtDNA replication results in quantitative alterations (reduction in mtDNA copy number, i.e., mtDNA depletion) and/or qualitative alterations (multiple deletions in mtDNA and, in some cases, point mutations). For this reason, this specific group of disorders is known as mitochondrial DNA depletion and multiple deletions syndromes (MDDS). These diseases are somehow peculiar because they respond to Mendelian inheritance, but the somatic aberrations (not inherited from the mother) in the mtDNA molecules confer them some molecular features typical of primary mtDNA disorders, such as different heteroplasmy degrees in different tissues.

Traditionally, mtDNA maintenance defects have been regarded as two different groups of disorders: mtDNA depletion syndromes, characterized by the reduction in mtDNA copy number in affected tissues, and commonly found in paediatric patients with variable and often severe phenotypes [12,13] and mtDNA multiple deletions syndromes characterized by the presence of this molecular aberration [12]. This second group is often defined by the most common clinical presentation associated with the presence of mtDNA multiple deletions in skeletal muscle (autosomal dominant or recessive progressive externa ophthalmoplegia, PEO) and is mainly found in adult patients [14] (https://pubmed.ncbi.nlm.nih.gov/29517884/, accessed on 22 May 2021). However, the knowledge acquired in recent years, as the genetic causes of these disorders are gradually being elucidated, indicates that both mtDNA depletion and multiple deletions may be the result of the same genetic defect and therefore primarily triggered by the same dysfunctional process (although likely at different developmental stages). Most genes associated with depletion in some patients have been found to cause multiple deletions in other patients, even when sharing the same mutation. In some cases, both molecular features may coexist in the same patient [15]. Although the biochemical mechanisms of both molecular alterations should be different, the primary genetic defect is common in many cases. Therefore, we refer to all this group of diseases commonly as MDDS.

The clinical presentation of the different MDDS forms is heterogeneous, even for groups of patients with mutations in the same gene. MDDS are associated with a wide phenotypic spectrum that affects different tissues, ranging from mild forms with presentation in adulthood to severe forms with infantile onset. Moreover, the mutations and phenotypes overlap, which means that alterations in one gene can lead to different phenotypes and similar phenotypes can be caused by mutations in different genes. All these factors make it difficult to associate a clear phenotype to these syndromes in some cases, and therefore to establish their diagnosis and classification.

Table 1 lists the genes whose mutations have been hitherto associated with MDDS. These genes can be classified into four categories according to the biological role of their products which inform us of the pathomechanism accounting for the mtDNA replication defect: (1) proteins belonging to the mtDNA replication machinery; (2) enzymes involved in deoxyribonucleoside triphosphates (dNTP) metabolism; (3) proteins involved in mitochondrial dynamics; and (4) other proteins whose link with mtDNA replication is unknown. In this last group, we find four mitochondrial membrane transporters, as well as other proteins with different functions whose link with mtDNA maintenance is not easily identifiable.

### 2.1. Genes Encoding Proteins of the mtDNA Replication Machinery

mtDNA replication is entirely dependent on nuclear encoded proteins that are synthesized in the cytosol and imported into mitochondria. Interestingly, mutations in genes encoding most of the factors constituting the mtDNA replisome [48,49] have been associated with MDDS (Table 1). Since the first identification of mutations in *POLG* [16] and *TWNK* [18] as causative of MDDS in 2001, a total of 10 genes encoding proteins directly involved in mtDNA replication and repair machinery have been incorporated in this list to date, including the recent identification of mutations in *LIG3* causing a mitochondrial neurogastrointestinal encephalomyopathy (MNGIE) like phenotype [25].

The first protein in this list is polymerase gamma, the mitochondrial replicase [50]. It is a heterotrimer constituted of a large subunit of 140 kDa encoded by *POLG* that has the catalytic activities (5′-3′ polymerase, 5′ deoxyribose phosphate lyase and proofreading 3′-5′ exonuclease), and a homodimeric subunit formed by two small peptides of 55 kDa, each encoded by *POLG2*, which acts as a DNA binding factor providing processivity to the complex [49]. Mutations in *POLG* are the most frequent genetic cause of MDDS (in fact, the main cause of mitochondrial diseases) [51]. More than 300 mutations affecting all the domains of the protein have been described in *POLG* (https://tools.niehs.nih.gov//polg/, accessed on 22 May 2021), affecting both polymerase and proofreading activities, and resulting in mtDNA depletion, multiple deletions and point mutations [51,52].

Mutations in *POLG* cause many different syndromes with variable severities (either paediatric or adult disorders, and with recessive or dominant inheritance). These include Alpers–Huttenlocher Syndrome (AHS), sensory ataxic neuropathy, disarthrya and ophthalmoparesis (SANDO), neurogaostrointestinal encephalomyopatic syndrome (MNGIE type), and dominant or recessive progressive external ophthalmoplegia (PEO). A recent study compiling 155 patients with mutations in *POLG* found that neurological, ophthalmological and gastrointestinal are the most frequently reported symptoms. Moreover, the symptoms are associated with the age of onset, with feeding difficulties and seizures being the most common symptoms in people under 12 years, and ataxia, peripheral neuropathy and seizures in patients between 12 and 40 years of age and ptosis, PEO and ataxia in patients with onset above 40 years [53]. Mutations in *POLG2*, encoding the ancillary subunit of polymerase gamma, are much rarer than those found in *POLG*, but since the number of patients with new pathogenic mutations in *POLG2* is increasing, it has become evident that the range of phenotypes is also variable [17,54], including fulminant cases [55,56].

Additionally, in 2001, mutations in *TWNK* (formerly called *C10orf2*) were associated with cases of autosomal dominant progressive external ophthalmoplegia (adPEO) with multiple mtDNA deletions [18]. This gene encodes twinkle, a homohexameric 5′-3′ DNA mitochondrial helicase that unwinds the double-stranded mtDNA exposing the single strands to the polymerase gamma to be used as a template for replication [57]. Mutations in *TWNK* found in adPEO patients with multiple deletions affect its helicase activity, stall the replication fork and cause mtDNA depletion in vitro, and also in some in vivo conditions (transgenic mice expressing adPEO variants) [58]. These observations support the notion that mtDNA depletion and multiple deletions are triggered by similar dysfunctional events. In fact, *TWNK* mutations also cause mtDNA depletion, associated with more severe phenotypes, such as infantile onset spinocerebellar ataxia (IOSCA) [59], or other phenotypic variants associated with mtDNA multiple deletions such as Perrault syndrome [60,61]. Interestingly, mtDNA somatic point mutations at the control region have been found in the skeletal muscle of patients with MDDS caused by mutations in *POLG* and *TWNK* [62], but to a lesser extent than those found in MNGIE patients caused by mutations in *TYMP* [63] (see below), which underlines the fact that the generation of somatic point mutations in mtDNA is another molecular feature of MDDS, although less prominent (and likely less functionally relevant) than depletion or multiple deletions.

As shown in Table 1, mutations in many other proteins belonging to the mtDNA replication/repair machinery have been gradually identified since 2013 [19,20,21,22,23,24,25,64,65], probably owing to the widespread introduction of deep sequencing methodology for genetics diagnostic. The group includes genes involved in mtDNA repair (*DNA2*, and possibly *MGME1*) and other long-ago known proteins participating in mtDNA replication such as mtSSB and TFAM. The range of different phenotypes associated with these disorders is also wide, from mild or moderate myopathy to severe phenotypes with multisystem involvement.

### 2.2. Genes Involved in dNTP Metabolism

The second group of genes associated with MDDS is constituted by genes encoding enzymes that catalyse reactions of dNTP metabolism. Since mtDNA needs a balanced supply of dNTPs for replication and repair, it is not surprising that mutations in these genes cause mtDNA alterations. Some of the MDDS that are currently treated with specific therapies belong to this group and their therapeutic approaches are targeted to modify/correct dNTP homeostasis (see Section 4.5 and Section 4.6 below). Therefore, we will pay special attention to dNTP metabolism in this review.

There are two anabolic pathways to obtain dNTPs in cells: the de novo pathway that is fully active in dividing cells only, and the salvage pathway that recycles deoxyribonucleosides from endogenous turnover and from the diet and takes place in both dividing and non-dividing cells. In addition, catabolism (consecutive dNTP dephosphorylation and further degradation of the resulting deoxyribonucleosides) also contributes to dNTP homeostasis. Alterations in both anabolic and catabolic pathways are involved in MDDS (Figure 1).

#### 2.2.1. dNTP Anabolism

##### De Novo Pathway

The main source of nucleotides (which includes dNTPs) is the de novo pathway, which takes place in the cytosol and is fully active in S phase. Small molecules such as amino acids and CO_2_ are used as substrate precursors to obtain nucleoside diphosphates (NDPs), which are then reduced to deoxyribonucleoside diphosphates (dNDPs) by the ribonucleotide reductase (RNR). RNR is a key enzyme in the de novo pathway that works as a heterotetramer composed of two copies of a large catalytic subunit (R1 encoded by *RRM1*) and two copies of a small subunit that can be presented in two isoforms (R2 encoded by *RRM2*, or p53R2 encoded by *RRM2B*). During S phase, the complex RNR is composed by 2R1/2R2 while in cells out of the S phase, the transcription of subunit R2 is inhibited, R2 subunits are degraded, and R1 thus solely interacts with p53R2, which is expressed throughout the cell cycle. The activity of the complex 2R1/2R2 is higher than 2R1/2p53R2 because in the S phase, the needs of dNTPs are much higher than in the rest of the cell cycle [66]. The dNDPs formed as products of RNR catalysis are then further phosphorylated by nucleoside diphosphate kinases (NDPKs) to dNTPs, the building blocks of DNA. There is biochemical evidence that deoxyribonucleotides can be imported from cytosol to mitochondria [67,68], and specific transporters for pyrimidine nucleotides have been identified [69,70,71]. The dNTP de novo pathway also depends on the enzyme thymidylate synthase (TS) that methylates deoxyuridine monophosphate (dUMP) using methylenetetrahydrofolate (MTHF) as a methyl donor, to generate thymidine monophosphate (dTMP). dTMP is further phosphorylated by thymidine monophosphate kinase (TMPK) and NDPK to obtain dTTP. Both TS and TMPK are cell cycle regulated and are mainly expressed in S phase [66].

Mutations in one gene belonging to the de novo pathway, *RRM2B*, have to date been associated with MDDS. As mentioned above, *RRM2B* encodes p53R2, the p53-regulated small subunit of RNR. Initially, mutations in this gene were associated with a fatal encephalomyopathic form of MDDS with renal tubulopathy and profound mtDNA depletion [29]; however, further cases have been associated with less severe phenotypes, such as MNGIE-like clinical presentations [72], or dominant or recessive PEO and PEO-plus associated with mtDNA multiple deletions [73,74,75,76,77].

##### Salvage Pathway

As indicated above, the dNTP salvage pathway recycles deoxyribonucleosides (dNs) derived from diet or from cellular metabolism (DNA degradation, de novo synthesis) to dNTPs. It takes place in both cytosol and mitochondria and it is not cell-cycle regulated. It relies on two parallel sets of enzymes in cytosol and mitochondria that sequentially phosphorylate dNs to dNTPs. The first phosphorylation (from dN to dNMP) is the rate-limiting step; in the cytosol, it is catalysed by thymidine kinase 1 (TK1), which phosphorylates thymidine, dThd (and also deoxyuridine, dUrd), and deoxycytidine kinase (dCK) that phosphorylates deoxycytidine (dCtd), deoxyadenosine (dAdo) and deoxyguanosine (dGuo). The equivalent enzymes in mitochondria are thymidine kinase 2 (TK2), which phosphorylates the pyrimidine dNs (dThd, dCtd and dUrd), and deoxyguanosine kinase (dGK), which phosphorylates the purine dNs (dAdo and dGuo). dNMPs produced in these first steps are further phosphorylated by nucleoside monophosphate kinases (NMPKs) and nucleoside diphosphate kinases (NDPKs), before finally obtaining dNTPs.

Mutations in *TK2* and *DGUOK* (encoding TK2 and dGK) are associated with severe MDDS [27,28]. Mutations in *TK2* cause a myopathic form with a considerable clinical variability, ranging from severe presentations that result as fatal in infancy or childhood, associated with marked mtDNA depletion in muscle, to less severe presentations that are usually associated with mtDNA multiple deletions, with slower clinical progression or even onset in adulthood [78]. *DGUOK* mutations cause severe hepatopathy or hepatocerebral syndrome with mtDNA depletion in infants, although myopathic cases with multiple deletions have also been reported [28,79,80].

#### 2.2.2. dNTP Catabolism

dNTPs are catabolized through consecutive dephosphorylating reactions (reverse to those described for the salvage pathway) and further degradation of the resulting dNs. Several specific enzymes catalyse some catabolic reactions biochemically different than those mediated by the anabolic enzymes, thus constituting independent regulatory catabolic steps. 5′-nucleotidases, with several isoforms including a mitochondrial isoform encoded by *NT5M* [81], catalyse the hydrolytic dephosphorylation of the monophosphates, and together with the anabolic (deoxy)ribonucleoside kinases, constitute substrate cycles that regulate cytosolic and mitochondrial monophosphate concentrations at the expense of ATP consumption [81]. SAMHD1, is a cytosolic triphosphohydrolase that catalyses the direct dephosphorylation of dNTPs to dNs [82], and dCTP pyrophosphatase 1 hydrolyses dNTPs to the corresponding monophosphates [83,84]. dN degradation is initiated by cytidine deaminase (CDA, conversion of dCtd to dUrd), adenosine deaminase (ADA, conversion of dAdo to deoxyinosine, dIno). The resulting dNs dUrd and dIno, together with the canonical dGuo and dThd, are broken by phosphorolysis by purine nucleoside phosphorylase (PNP, for dGuo and dIno) and thymidine phosphorylase (TP, for dThd and dUrd).

The latter enzyme, TP, is encoded by *TYMP*, and it is the only nucleotide/nucleoside catabolic enzyme associated with MDDS. Mutations in *TYMP* cause mitochondrial neurogastrointestinal encephalomyopathy (MNGIE), a recessive disorder characterized by gastrointestinal dismotility, cachexia, ptosis, progressive external ophtalmoplegia, peripheral neuropathy and leukoencephalopathy [15,26,85]. As a consequence of TP dysfunction, MNGIE patients present elevated systemic concentrations of dThd and dUrd. Experimental results show that this dN overload results in increased dTTP levels and secondary dCTP depletion in mitochondria, which interferes with mtDNA replication and maintenance, causing secondary mtDNA depletion in vitro and in organello [86,87,88,89]. In vivo results obtained in a genetic murine model of the disease confirmed these biochemical disarrangements [90]. As a consequence of this dNTP imbalance, MNGIE patients present mtDNA depletion, multiple deletions and somatic point mutations [15,63,91]. As the toxic effect of dThd and dUrd on mtDNA is the biochemical imbalance triggering the phenotype in MNGIE, most therapy approaches aim to eliminate these nucleosides using different strategies (see Section 4 below).

The de novo pathway is active in replicating cells, providing elevated dNTP cellular levels needed for nuclear DNA replication. Post-mitotic cells downregulate the de novo pathway, have reduced dNTP levels, and thus become much more dependent on the cytosolic and mitochondrial salvage pathways. In the particular case of the dThd, the cytosolic salvage pathway is also inhibited, because TK1 is downregulated. Therefore, the salvage of dThd mainly depends on the mitochondrial TK2 in quiescent cells [68].

#### 2.2.3. Nucleoside/Nucleotide Transporters

As mentioned above, biochemical evidence indicates that deoxyribonucleotides can be exchanged between cytosol and mitochondria through the inner mitochondrial membrane [67,68], and some of the mitochondrial transporters have been identified [69,70,71]. Although mitochondrial and cytosolic dNTP pools are different, they are not mutually independent, and deoxyribonucleotide transport across the mitochondrial membrane contributes to keep a balanced dNTP composition in both compartments. Mitochondrial import likely predominates in dividing cells, while there is also evidence of mitochondrial export in quiescent state, at least for the deoxyribonucleotides of thymidine [67].

On the other hand, plasma membranes are impermeable to nucleotides because they do not have transporters enabling the internalization of these charged compounds [92]. Instead, nucleoside and nucleobase transport between the cell and the extracellular milieu is mediated by two families of carriers, with different isoforms having different substrate selectivity: the solute carrier family 28 (*SLC28* genes) encoding the concentrative nucleoside transporters CNT (CNT1-3), and the solute carrier family 29 (*SLC29* genes) encoding the equilibrative nucleoside transporters ENT (ENT1-4). Concentrative and equilibrative transporters are located in the plasma membranes of cells, with different distribution in different cells and even different regions of the plasma membrane to promote vectorial nucleoside transport through the cell [93,94]. Thus far, only one of these transporters, ENT1, has been found in the mitochondrial membrane [95].

Although mutations in some nucleoside transporters may be pathogenic [96,97], to date, no mitochondrial diseases have been associated with mutations in these genes. However, these transporters are very important in biomedicine, as they are obligated mediators for some nucleotide-based therapies, such as chemotherapy for cancer or some antivirals [93], and they should also be involved in some of the emerging therapies for MDDS discussed in Section 4 below.

### 2.3. Genes Involved in Mitochondrial Dynamics

A functional mitochondrial dynamics is essential for mtDNA replication because the fission and fusion of the organelle allow mitochondria to exchange components of the matrix, thus facilitating a balanced composition of the mitochondrial dNTP pool and mtDNA replication enzymes, as well as an even distribution of mtDNA molecules throughout the cellular mitochondrial network [98,99].

Mitochondria mix through the coordinated fusion of the outer and inner membranes by the action of three GTPases: mitofusin 1 (MFN1) and 2 (MFN2) mediate the fusion of the outer mitochondrial membrane while the dynamin-related protein OPA1 is essential for inner membrane fusion [99]. In addition, *OPA1* encodes an alternatively spliced isoform, OPA1-exon4b, that is associated with the mitochondrial inner membrane and directly interacts with the nucleoids, allowing their distribution within the mitochondrial network. OPA1-exon4b also promotes mtDNA replication by interacting with and regulating the replisome [100]. Thus, it is not surprising that dysfunctional mitochondrial dynamics interferes with the correct replication of mtDNA, although the details of the molecular mechanisms accounting for the reduction in the mtDNA copy number or the occurrence of multiple deletions may be different for mutations in different genes of this category, and so these molecular details need to be specifically explored for each gene.

Mutations in *MFN2* and *OPA1* cause MDDS, presenting as optic atrophy, myopathy, axonal neuropathy and Charcot–Marie–Tooth disease in the case of *MFN2* [31] and optic atrophy and Behr syndrome in the case of *OPA1* [30]. Protein misato homolog 1, encoded by *MSTO1*, is associated with the mitochondrial outer membrane and participates in mitochondrial distribution and network formation [101]. Mutations in *MSTO1* cause myopathy and ataxia [34]. *MICOS13* encodes MIC13, a subunit of the mitochondrial contact site and cristae junction organizing system (MICOS). Mutations in this gene cause liver mtDNA depletion, OXPHOS deficiency and fewer cristae structures [35,102]. Paraplegin, encoded by *SPG7*, and AFG3L2, are components of the m-AAA protease complex, located in the inner mitochondrial membrane. This proteolytic complex degrades misfolded proteins and regulates ribosome assembly [32,103]. Recently, it has been described that mutations in *AFG3L2* destabilize OPA1 leading to mitochondrial fragmentation [104]. Mutations in *SPG7* and *AFG3L2* have been associated with PEO accompanied by movement disorders with mtDNA multiple deletions in muscle [32,33,105].

### 2.4. Genes Involved in mtDNA Maintenance through Unknown Mechanisms

Approximately one third of all genes associated with MDDS to date encode proteins that do not have a clear functional link with mtDNA replication. This group constitutes the last category of genes listed in Table 1. Some classifications include several of these genes (*SLC25A4*, *AGK*, *SUCLA2*, *SUCLG1*, *ABAT*, *MPV17*) as related with nucleotide metabolism [106], however, we prefer to restrict this group to genes that directly participate in dNTP homeostasis, as these are the actual substrates of mtDNA replication. *SLC25A4* encodes ANT1, the muscle isoform of the ADP/ATP mitochondrial translocator (adenine nucleotide translocator, ANT). ATP and ADP nucleotides are not mtDNA building blocks, therefore, the reason why mutations in this gene produce mtDNA multiple deletions seems not to be directly related with dNTP imbalance. Mutations in this gene were associated with myopathy with mtDNA multiple deletions as early as 2000 [36].

Similarly, mutations in *AGK* have been associated with mtDNA depletion in Sengers syndrome [42,107]. This gene encodes the enzyme acylglycerol kinase that phosphorylates monoacylglycerol and diacylglycerol to phosphatidic acid, an intermediate to synthesize membrane lipids such as cardiolipin. Since this lipid is associated with ANT [108], it has been hypothesized that AGK could play a role in nucleotide import via ANT by regulating the lipid composition of the inner mitochondrial membrane, and thus this gene is sometimes classified in the group of dNTP-related genes. However, as pointed out above, ANT does not regulate dNTPs and there is no obvious reason to assume that it participates in dNTP homeostasis. Mutations in *SUCLA2* and *SUCLG1* have been associated with encephalopathy with mtDNA depletion [40,41]. These genes encode two subunits of Krebs cycle enzyme succinate CoA ligase. Although this protein is a molecular partner of the mitochondrial isoform of the dNTP anabolic enzyme nucleoside diphosphate kinase (NDPK) [109], the assumption that mutations in *SUCLA2* and *SUCLG1* disrupt dNTP homeostasis because of this molecular association has not been supported by experimental data. *ABAT* encodes the mitochondrial enzyme gamma-aminobutyrate aminotransferase, which catalyses the catabolic deamination of the neurotransmitter gamma-aminobutyric acid (GABA). The main reason that this enzyme has been claimed to have a role in dNTP homeostasis relies on the observation that dNTP addition rescues mtDNA depletion, as observed in *ABAT*-deficient cultured cells [44]. In fact, dNs are the actual molecules that enter the cell after dNTP addition to the cell culture [92], and the enhancement of mtDNA replication by intracellular dN-stimulated dNTP synthesis has been observed for deficiencies other than those directly related with dNTP synthesis [110,111]. In fact, increased dNTP availability may increase mtDNA copy number even in fully functional cells [112]. Similarly, although dN supplementation can rescue mtDNA depletion in *MPV17* mutants in vitro and in vivo [111], it has not yet been demonstrated that the membrane protein transporter MPV17 (associated in patients with severe neurohepatopathy with mtDNA depletion [37]) has a role as a nucleotide carrier. Therefore, the biochemical mechanisms accounting for the defective mtDNA replication/maintenance observed in patients with mutations in *SLC25A4*, *AGK*, *SUCLA2*, *SUCLG1*, *ABAT* and *MPV17* remain to be clarified.

The biological function of the proteins included in this last category is variable, and the severity and phenotype features of the patients is as diverse as those observed in the other categories. Interestingly, four genes of this group encode transporters located in the inner mitochondrial membrane, functioning as carriers of different molecules. In addition to the aforementioned carriers ANT1 and MPV17, mutations in two additional genes (*SLC25A21*, *SLC25A10*) encoding two mitochondrial dicarboxylate transporters have been found in patients with mtDNA depletion [38,39]. The other genes of this group include GFER and FBXL4. *GFER* encodes a growth factor whose mutations cause myopathy, congenital cataract and developmental delay. In vitro, mutations in *GFER* are associated with mtDNA multiple deletions in patients’ myoblasts but not in fibroblasts [43,113]. *FBXL4* encodes a protein located in the mitochondrial intermembrane space that belongs to the F-box family, whose exact function is not yet known. Although proteins of this family participate in Skp1-cullin1-F-box (SCF)-dependent proteasomal ubiquitin-dependent protein catabolic process, FBXL4 has also been associated with mitochondrial dynamics [106]. However, studies conducted in *Fbxl4* knockout mice have revealed that the molecular phenotype caused by *FBXL4* mutations (mtDNA depletion observed in patients [45]) may be mediated by a global decrease in mitochondrial content due to the dysregulation of autophagy rather than a primary dysfunctional mtDNA maintenance [114]. C1QBP (complement component 1 Q subcomponent-binding protein) is a mitochondrial matrix protein of unknown function which is thought to be involved in inflammation and mitochondrial ribosome synthesis. Patients with a clinic of PEO carrying mutations in this gene showed mtDNA multiple deletions in muscle but not depletion [47,115].

Three of the genes quoted at the OMIM database were associated with mtDNA depletion (OMIM: Phenotypic Series—PS603041, mitochondrial DNA depletion syndrome) have been reported in isolated single cases to date (*SCL25A21* [38], *SCL25A10* [39] and *MRM2* [46]), and for two of these, the reported degree of mtDNA depletion is mild (*SCL25A10* and *MRM2*). It is expected that new cases of MDDS patients with mutations in these genes will confirm their association with mtDNA replication disorders.

## 3. Non-Targeted Therapies for Mitochondrial DNA Maintenance Disorders

MDDS are mitochondrial disorders. The OXPHOS dysfunction derived from the somatic mtDNA alteration is the factor causing the clinical presentation in MDDS patients. As with other mitochondrial diseases, treatments conceived to ameliorate mitochondrial dysfunction are often applied to MDDS patients even when acting through non-targeted or unspecific mechanisms, and some of the experimental approaches under preclinical investigation have also been explored in MDDS models.

### 3.1. Symptomatic Treatments and Other Non-Targeted Treatments Applied to MDDS Patients

MDDS are diseases with poor prognosis in the majority of patients and there is currently no curative therapy for any of them [116]. Therefore, with the exception of some isolated cases in which investigational targeted treatments are applied under compassionate use or have initiated a clinical trial, current treatments focus on mitigating the symptoms and preventing further complications. The main advantage of these treatments is that, in principle, they can be applied to all MDDS patients regardless of the genetic cause, although their real efficacy or impact on the disease can be mild due to their generalist mechanism of action.

Seizures is one of the serious manifestations that may occur in MDDS patients (observed in some forms caused by mutations in *POLG*, *TWNK* and others [117]). They can be treated with antiepileptic drugs, but valproate is not recommended, because mitochondrial dysfunction is a risk factor for valproate-induced liver failure [118]. Nasogastric tube or gastrostomy tube feedings are ways to treat the feeding difficulties and failure to thrive. Chest physiotherapy, artificial ventilation, tracheostomy or ventilators can be used to treat respiratory insufficiency. Other treatments include the surgical treatment of scholiosis, kyphosis, or ptosis and cochlear implantation in the cases of hearing loss [119].

Special diets are also a possibility used in some mitochondrial diseases [116,120]. Options such as continuous feeding in order to prevent hypoglycaemia [119], or the administration of ketogenic high-fat low carbohydrate diet to stimulate beta-oxidation, can be recommended depending on the specific presentation of the patient. Ketogenic diet reduced the heteroplasmic load in cybrids carrying mtDNA single deletion [121] and slowed down myopathy progression in mice [122].

Supplementation with key compounds of the ETC is a widely used treatment for MDDS. For instance, succinate and ubiquinone have been used to slow down liver dysfunction in *MPV17* patients [123], and treatments with oral folinic acid [124], levocarnitine, carnitine, coenzyme Q10, idibenone, alpha lipoic acid, and vitamins C, E and B are also used, often together as a cocktail [125]. Nevertheless, the evidence of their effectiveness is short [126] and there are no standard guides recommending any specific cocktail of compounds.

Physical therapy can help maintain muscle tone and function [119]. Moreover, it has been described that exercise stimulates mitochondrial biogenesis and thus counteracts the oxidative dysfunction [127]. Additionally, a published review of the literature recommends endurance and resistance exercise [128].

### 3.2. Non-Targeted Experimental Approaches

Several non-targeted experimental strategies have been proposed to fight mitochondrial dysfunction, such as the activation of mitochondrial biogenesis to compensate ATP depletion [129], the inhibition of mTORC1, which is an activator of anabolic pathways and an inhibitor of autophagy [130], hypoxia [131] or bypassing the OXPHOS defects with non-mammalian alternative oxidase expression [132]. None of these strategies have been tested in any preclinical model of MDDS, with the exception of mTORC1 inhibition, which has been tested in a TK2 knock in mouse model of MDDS. Low-dose rapamycin extended the lifespan of the TK2-deficient mice by nearly two-fold. Interestingly, the treatment did not rescue mtDNA depletion, and did not prevent mitochondrial dysfunction but induced metabolic changes, likely promoting the use of alternative energy reserves and a developmental reprogramming that ultimately prolongs survival [133].

Treatments addressing primary mutations in mtDNA, such as mitochondrial replacement therapy [134] or mtDNA heteroplasmy shifting [135], could hardly be applied to MDDS patients.

## 4. Targeted Therapies for MDDS

The first patients with mtDNA depletion and multiple deletions were reported long ago [136,137], and the first genetic cause for one of the MDDS was found in 1999 [26]. The significant advances in our knowledge of the biochemical and molecular pathomechanisms accounting for dysfunctional mtDNA replication in many of these disorders, gained over the two last decades, have opened the possibility to design and study tailored strategies to fight the specific cause of each form of MDDS. Although these therapies are normally designed to specifically address a particular defect, they can, in some cases be extended to disorders with a different aetiology but a common underlying biochemical or molecular defect, likely being more effective than general non-targeted treatments. Some of these approaches are still in the preclinical phases, but in some cases, they have already reached the clinical trial phase (Table 2).

### 4.1. Direct Scavenging of Toxic Metabolites

Although there are other mitochondrial diseases caused by the toxic effect of accumulated metabolites [138], mitochondrial neurogastrointestinal encephalomyopathy (MNGIE) is the only disease falling in the category of MDDS, to date identified as caused by this type of biochemical noxious effect. In MNGIE, mtDNA replication is interfered by the dNTP imbalances caused by the systemic accumulation of dThd and dUrd, which are the substrates of the dysfunctional enzyme thymidine phosphorylase [87,89]. Therefore, most therapy strategies proposed for MNGIE patients, including allogeneic hematopoietic stem cell transplantation (AHSCT) or liver transplantation, aim at clearing the biochemical excess of these compounds (see below).

Since dThd and dUrd are filterable water-soluble molecules, the first therapeutic approach for MNGIE was haemodialysis. However, this method was ineffective because the reduction in circulating dThd levels was suboptimal (not reaching normal values) and transient. Three hours after the dialysis, plasma dThd levels returned to their initial values, even after several consecutive repeated rounds of dialysis [89]. More recent results showed that haemodialysis does not remove excessive metabolites from the cerebrospinal fluid (CSF) [139]. Likewise, continuous ambulatory peritoneal dialysis has been attempted but its effects were also transient [140]. Peritoneal dialysis has been proposed as a bridge or maintenance treatment for patients while they wait for a compatible donor for AHSCT or liver transplantation [85].

The results obtained with haemodialysis in MNGIE patients revealed that urine dThd is reabsorbed in the kidney, so it was proposed to inhibit this reabsorption to contribute to eliminate this metabolite from patients, but finding a drug that selectively inhibits the reabsorption of pyrimidine dNs without affecting the reabsorption of other compounds is difficult [141].

Regarding toxic metabolite accumulation in MDDS other than MNGIE, the existence of an oxidative environment favours the oxidation of cysteine to cystine, which accumulates in several tissues mainly affecting kidneys and eyes. The use of cysteamine bitartrate that catalyses the conversion of cystine to cysteine, was proposed to prevent the formation of cystine crystals. Two clinical trials assessed its efficacy in MDDS (NCT02023866 and NCT02473445, Table 2), but they were prematurely closed because of the lack of efficacy they demonstrated.

### 4.2. Enzyme Replacement

Again, enzyme replacement has only been explored for MNGIE among MDDS. The dysfunctional enzyme in this disease, TP, is highly expressed in platelets and white blood cells [142,143] where it clears the metabolites dThd and dUrd, thus contributing to the removal of these nucleosides from all organs. As these nucleosides freely equilibrate across plasma membranes through dedicated equilibrative or concentrative transporters [93,144], clearance from plasma also contributes to their clearance from the tissular intracellular space. Two different strategies were designed to restore circulating TP activity in MNGIE patients, but only one of them has reached the use in patients.

The first one was based on the use of polymeric nanoreactors [145,146]. Preclinical in vitro and in vivo experiments showed that TP-loaded nanoreactors were enzymatically active and stable in blood serum, and they did not induce inflammatory response after intraperitoneal administration in mice [145,146]. However, no further advances on this strategy have been reported in recent years.

The second enzyme replacement strategy for MNGIE proposes the use of erythrocyte entrapped thymidine phosphorylase (EETP). This approach is based on the ex vivo encapsulation of the recombinant TP enzyme within patient’s erythrocytes to be reinfused into the patient [147]. This strategy prevents immune response to exogenous TP [148]. Erythrocyte membranes are permeable to dNs, so recombinant TP can catabolize internalized metabolites and thus reduce dN blood levels. The reduction in dThd and dUrd levels achieved in MNGIE patients by EETP is suboptimal and transient (blood valley levels after infusion of EETP to patients do not reach normal levels), and infusion should be periodically repeated every few weeks [147,149,150]. On the other hand, the reports indicate that this treatment improved the condition of the patients [149,150]. Like peritoneal dialysis, EETP can be used as a bridge therapy for MNGIE but perhaps it is not a feasible option as a definitive treatment. Currently, there is a clinical trial assessing the efficacy of this therapy [151] (see Table 2).

### 4.3. Allogeneic Hematopoietic Stem Cell Transplantation (AHSCT)

As mentioned above, platelets and white blood cells are among the richest reservoirs of TP in humans [142,143]. This cytosolic enzyme is not excreted and exerts its catabolic role intracellularly. Substrates (dThd and dUrd) from the extracellular compartment are internalized into the cell, where TP catalyses their phosphorolysis and then the reaction products can be further catabolized. In this way, white blood cells and platelets (as other tissues highly expressing TP, such as the liver) act as “systemic clearing reactors” for dThd and dUrd. Considering this physiological role of platelets and white cells, the possibility to treat MNGIE by AHSCT was considered.

To gain a first proof of concept for this approach, the effect of platelet infusions from healthy donors was studied in vitro and in MNGIE patients. Patients treated with this strategy reduced dThd and dUrd levels in blood, but this reduction was transient and suboptimal due to the short half-life of platelets (8–9 days) [152]. In fact, platelet infusion was not initially conceived as an actual therapy for patients (platelets would have to be infused in patients during the whole life), but as a test to prove that AHSCT would be effective in removing circulating dThd and dUrd from patients [152]. Nevertheless, some authors have tried to use platelets in other patients with similar results [153].

The first report of two patients treated with AHSCT confirmed that, after successful engraftment, circulating levels of dThd and dUrd were lowered to undetectable or barely detectable levels [154]. However, the requirement of a compatible donor, together with the need for a toxic conditioning regime before the transplantation and the potential severe adverse effects of AHSCT, make it a complex procedure associated with relevant risks. Consensus recommendations were published based on the experience collected from several cases of AHSCT in MNGIE [155]. The results of this cohort constituted a retrospective study of 24 patients treated with AHSCT, and revealed that only nine patients were alive in the last follow-up and among the 15 patients that did not survive, nine had died due to transplant-related causes while six died due to complications from MNGIE [156]. Nonetheless, there was an improvement in the body mass index, gastrointestinal manifestations and the peripheral neuropathy in the survivors, although all these positive effects were very slow and became objectively confirmed only after several years of follow up. In order to maximize the success of this intervention, AHSCT should be considered only for selected patients that are relatively healthy without severe gastrointestinal or liver affectation [157]. A recent position paper contains updated recommendations that help decide in what MNGIE cases liver transplantation (see below) is a preferable choice than AHSCT [85]. An ongoing clinical trial is studying the efficacy and safety of this treatment (see Table 2).

### 4.4. Liver Transplantation

The liver is a severely affected organ in some forms of MDDS such as those caused by mutations in *DGUOK*, *MPV17* and others. Liver transplantation is an option for these patients, although limited to cases of isolated liver involvement, because the multi-organ affectation would not be corrected with this intervention [158]. Some patients with mutations in *MPV17* have been treated with this strategy, but its efficacy is controversial because more than half of the transplanted children died in the post-transplantation period [159] (https://pubmed.ncbi.nlm.nih.gov/22593919/ accessed on 22 May 2021). A recent work has reported that liver transplant is only effective in patients with mutations in *MPV17* with late onset and mild phenotype [160]. Liver transplantation has also been used in *DGUOK* patients since most of them show severe hepatic dysfunction. The results have been highly uneven, with some transplanted patients surviving longer than 5 years, while in other cases non-transplanted patients had a better outcome [161]. Additionally, patients with mutations in *POLG* can benefit from this treatment, but only if they do not manifest Alpers–Huttenlocher syndrome, because transplantation does not revert the brain alterations [162]. In one report, liver transplantation has been applied to two adults with mutations in *POLG*, but only one of them survived the procedure [163]. Another reported patient lived more than 20 years after liver transplantation (following valproate-induced liver failure) with semi-independent life [164].

Liver transplantation was also proposed [165] and has been applied [166,167,168,169] in MNGIE patients. In this case, the rationale of the treatment is different. Although it is not infrequent to observe mild liver affectation in some MNGIE patients [15], cases of severe liver dysfunction have not been reported. The objective of liver transplantation in MNGIE patients is not substituting a severely damaged organ, but to use the healthy liver as a source of “enzyme replacement” of the defective TP enzyme. The liver is very rich in TP [143] and the objective of the transplantation is that the grafted healthy organ clears the accumulated toxic nucleosides on a systemic level, similarly to what is pursued with AHSCT (see Section 4.3). This treatment has been reported in six patients and all of them were alive at the time of publication [166,167,168,169]. At least two more MNGIE patients have been liver transplanted and are alive 2 years after the treatment (unreported cases treated in our centre). Although the number of patients treated with liver transplantation is lower than those treated with AHSCT, liver transplantation seems to be safer because the survival rate is better and there is no aggressive conditioning before the intervention [125]. In fact, AHSCT and liver transplantation are the only choices currently available for the permanent treatment of MNGIE, and recommending either of these two alternate treatments may depend on several patient-related factors, as discussed and advised in a recent consensus document [85].

### 4.5. Administration of Deoxyribonucleosides (dNs)

Four genes associated with MDDS encode enzymes catalysing dNTP anabolic (*TK2*, *DGUOK*, *RRM2B*) or catabolic (*TYMP*) reactions (Table 1). Some time ago, in vitro evidence showed that supplementation with dAMP and dGMP prevented mtDNA depletion in cultured fibroblasts from patients with mutations in *DGUOK* [170]. The effect of this treatment is mediated by the action of dAdo and dGuo, generated by ectonucleotidases outside the cell from the added compounds dAMP and dGMP, as charged compounds cannot enter the cells without dedicated transporters. In fact, the same effect on mtDNA was observed when adding dGuo alone to *DGUOK* mutant cells [92].

The first in vivo evidence further supporting the efficacy of this strategy was obtained in a TK2 p.H126N knock in model that recapitulates some clinical features of the disease and have a median survival of around 14 days [171]. TK2 catalyses the first phosphorylation of the pyrimidine dNs, dThd and dCtd. As a consequence of TK2 dysfunction, TK2 mutant mice present a reduction in mitochondrial dTTP and dCTP, leading to mtDNA depletion [171]. TK2 deficiency, in its most severe form, causes a fatal myopathic disease in early infancy [78]. There is a considerable variability in the clinical presentation, with mtDNA depletion associated with the most severe cases and multiple mtDNA deletions being more common in milder and adult cases [172], although the clinical severity does not correlate with the residual TK2 activity [173].

Treatment of the p.H126N knock in mice with dTMP and dCMP showed delayed disease onset, improved biochemical abnormalities, increased levels of mtDNA and doubled life span, which constituted the first evidence that this therapy could be effective in patients [174] and encouraged the first compassionate treatment of patients with TK2 deficiency. Further studies demonstrated, as previously showed in vitro with *DGUOK* mutants, that the active compounds are the unphosphorylated molecules dThd and dCtd, rather than the charged nucleotides dTMP and dCMP, which cannot enter the cells. Knock in mice treated with dThd and dCtd experienced the same effect on survival and other molecular variables as those observed when treated with the monophosphates [175]. These results were replicated in a TK2 knockout model of the disease [112,176]. Thus, the therapeutic effect occurs regardless of the presence of any residual TK2 activity, should be mediated by the parallel cytosolic salvage enzymes TK1 and dCK and is limited by age-dependent factors that may account for the apparent loss of effect in older animals [112,177].

The enzyme TK2 catalyses the phosphorylation of dThd and dCtd within mitochondria, resulting in the production of the corresponding monophosphates dTMP and dCMP (Figure 1). The mechanism accounting for the effect of dThd and dCtd in the aforementioned animal models is not obvious because these lack TK2 activity, so they still should be unable to phosphorylate these administered compounds with their dysfunctional TK2 enzyme. The biochemical reason accounting for this effect should be mediated by the cytosolic enzymes TK1 (that phosphorylates dThd) and dCK (that phosphorylates dCtd). In untreated animals, endogenous dThd and dCtd concentrations would not suffice to compensate for TK2 deficiency. However, high doses of dThd and dCtd administered to the animals may enhance the cytosolic salvage pathway through two overlapping mechanisms: (1) increased substrates will increase the saturation of TK1 and dCK, thus enhancing their catalytic activity; and (2) increased substrates will also displace the thermodynamics equilibrium of the cytosolic deoxypyrimidine salvage pathway, favouring the reactions towards the phosphorylated species. Once phosphorylated, they can enter mitochondria, bypassing the defective TK2 step (Figure 1). This can be considered a “substrate enhancement therapy” as named in the original report [175].

Therapy with dThd and dCtd was initiated under compassionate use in an initial cohort of 16 paediatric and adult patients from five different countries (400 mg/kg/day of each dN in most patients, lower doses in a few patients), and the results of the follow-up of this cohort showed that the treatment dramatically changed the course of the disease [178]. The subgroup of five treated patients with early onset and severe disease were alive at the time of the report, after several months/years of treatment, while in the untreated historical control group with early onset severe myopathy, only 27% of patients survived beyond 2 years after diagnosis [78]. Motor function and other clinical measures stabilized or improved in all treated patients [178]. Importantly, the treatment was not associated with serious adverse effects and although it provoked diarrhoea in some patients, this in no case required withdrawal of treatment. Mild elevation of transaminases was observed in two patients belonging to an expanded cohort, which normalized when the treatment was discontinued [178]. A study focused on the effect of dCtd/dThd on the respiratory function has recently been published, revealing improvement in several endpoints and a decreased number of respiratory infections [179]. Some clinical trials are currently assessing the safety and efficacy of the treatment with dCtd/dThd in TK2 deficiency (see Table 2). The existence of good clinical endpoints, as well as good biochemical biomarkers of the response to the treatment [180], will facilitate the evaluation of the results.

There is preclinical evidence obtained in vitro that this strategy enhances mtDNA replication in other diseases caused by anabolic defects in dNTP metabolism. Several studies have confirmed this result in *DGUOK* cells [92,170,181] as well as in cells with mutations in *RRM2B*, encoding the p53R2 subunit of the de novo enzyme ribonucleotide reductase [182]. In the case of MNGIE, which is caused by mutations in *TYMP*, encoding the catabolic enzyme TP, it has been suggested that the treatment with dCtd could also be considered, because mtDNA depletion seems to be the consequence of secondary dCTP depletion as a consequence of dThd excess [87].

Interestingly, similar positive effects of dN supplementation on mtDNA replication have been observed in vitro and in vivo for mutations in genes not directly related with dNTP metabolism, such as *POLG* and *MPV17* [110,111,181]. This is particularly significant for the case of patients with mutations in *POLG*, as this is the most prevalent gene among all those causing MDDS. In this particular case, it was verified that dN supplementation (which changes dNTP composition) does not induce somatic mutations in de novo synthesized mtDNA [110]. The biochemical mechanism accounting for the dN-induced enhancement of mtDNA replication, even with normal dNTP anabolism, may be mediated by an enhanced activity of polymerase gamma, which is activated by increased substrate (dNTP) availability.

In any case, further in vivo preclinical testing is needed to confirm the efficacy of dNs as a treatment for MDDS other than TK2 deficiency. Some existing animal models are not adequate for this purpose. For example, the mutator *Polg* knock in mouse has a dysfunctional proof-reading exonuclease activity and does not present mtDNA depletion [183], and it is thus not a good model to test whether dN administration enhances the polymerase activity of this enzyme. A recently generated *Polg^A449T/A449T^* mouse [184] (equivalent to the frequent A467T *POLG* mutation in patients) could be a more adequate model to test this therapy, although its phenotype is also limited (mainly molecular alterations).

### 4.6. Gene Therapy

In contrast to many of the above discussed treatments, gene therapy has been investigated for some forms of MDDS, only at the preclinical level. The most frequently used tool for gene delivery has been the adenoassociated virus (AAV) vectors. AAV are preferably used over other viral vectors because of the low risk of random insertion into nuclear genome as they are episomic, and because of their long-term persistence in cells [185]. However, the lack of genome integration results in a reduction in the transgene copy number because of the dilution effect in proliferating tissues. Another disadvantage of this strategy is the limited length of the cloned gene (less than 4.7 kb) and the difficulty to target some of the tissues affected in MDDS.

The first preclinical study of gene therapy for a MDDS was conducted in 2005, using a knockout murine model of the myopathy caused by mutations in *SLC24A4*, encoding the heart/muscle isoform of the mitochondrial adenine nucleotide translocator (ANT1). AVV vectors carrying the wild-type version of *Slc25A4* were generated. After some in vitro tests showing the effects of AAV transduction on mutant myoblasts obtained from the knockout mice, the vectors were injected in the skeletal muscle of neonatal mice, which increased *Ant1* expression and partially reverted the OXPHOS dysfunction, as assessed by histochemical and immunohistochemical methods [186].

AAV-mediated gene therapy has been also tested in a model of the hepatocerebral form of MDDS caused by mutations in *MPV17*. This study faced some difficulties derived from the frequently observed fact that genetically modified mice do not fully recapitulate the disease they are intended to model. Although the *Mpv17* knockout mouse shows many molecular features that recapitulate those observed in the human disease (the most relevant one being mtDNA depletion in the liver), the mice fail to develop any functional phenotype other than late-onset kidney dysfunction [187]. However, the authors successfully induced a phenotype resembling the hepatic dysfunction observed in patients, by feeding the animals a ketogenic diet. Under these conditions, *Mpv17* knockout mice developed severe liver cirrhosis and liver failure after a few weeks. The authors treated *Mpv17* knockout mice with AAV2/8 carrying the correct human coding sequence of *MPV17* by intravenous injection, and the treatment rescued mtDNA levels and OXPHOS function, and prevented the liver phenotype induced by the ketogenic diet [188].

Gene therapy has been studied for MNGIE. For this disease, targeting a specific tissue is not the main objective because, as previously mentioned, the pursued effect is to clear toxic nucleosides on a systemic level. Whatever organ or cells accessible to the blood stream may be a good target to transduce the clearing enzyme TP. As indicated for the murine *Mpv17* knockout model, the one available model for MNGIE (the double knockout for *Tymp* and *Upp1*) only recapitulates the biochemical imbalances of the disease [90].

Two different strategies have been tested. The first one targeted the hematopoietic tissue, which is one of the richest sources of TP in humans (but not in mice) using lentiviral vectors. This approach demonstrated a reduction in dThd and dUrd levels observed in the animal model to barely detectable levels [189]. The effect was sustained over the whole life of the animals [190]. An additional study provided evidence that this treatment results in phenotypic correction beyond the biochemical homeostasis restoration [191]. Importantly, integrational oncogenesis or the clonal expansion of the transduced cells was not observed [191].

However, this strategy was associated with increased mortality of the animals, most likely owing to the aggressive conditioning of recipient mice [190,191]. Therefore, a second strategy targeting the liver (again, a highly TP-expressing tissue in humans, but not in mice) with AAV vectors was tested, showing a long-term sustained and very efficient correction of the biochemical imbalance associated with the disease with only a mild dilution effect [192,193]. The comparison of different promoters showed that the one of the alpha-1-antitrypsin gene provided the best efficacy as compared to other liver or constitutive promoters [194]. Recently, the use of an enhanced animal model showed that this treatment also has a therapeutic effect on the neuromotor function of the mice [195].

All gene therapy approaches discussed above have shown very promising results. Unfortunately, there are several big steps to take before they can be translated to patients. One of the main limitations is the amount of vector needed for transduction in humans. In the case of the AAV vectors targeting the liver, we also need to consider the fact that the transduction efficiency is lower in humans as compared to that observed in mice [196,197,198]. The production of clinical grade vectors is expensive. Thus, the cost of producing vectors for the first preclinical regulatory safety studies and subsequent clinical trials to test the gene therapy treatments will require considerable investments that sometimes are hard to gather for rare diseases like MDDS. However, the potential societal benefits derived from the implementation of gene therapy for one or more forms of MDDS are beyond any doubt, which should encourage stakeholders to pursue this objective.

### 4.7. Improving Mitochondrial Shape and Other Approaches

As some MDDS are caused by mutations in factors regulating mitochondrial dynamics, strategies aimed at restoring these processes and improving mitochondrial shape have been considered. Moderate overexpression of *OPA1* is found to regulate the organization of respiratory complexes in mice with mutations in genes not related to MDDS [199]. Additionally, the inhibition of OMA1, which cleaves and degrades OPA1 under stress conditions, resulted in the rescue of mtDNA levels (but not respiratory function) in a non-related MDDS gene KO mouse [200]. Although these results show promising results, they should be tested in MDDS models.

MDDS caused by mutations in *MFN2* can be potentially treated with a strategy that uses a small mitofusin agonist that activates MFN2, thus promoting mitochondrial fusion [201]. This work showed that axonal mitochondrial trafficking, which is dysfunctional in Charcot–Marie–Tooth disease, was restored in treated *MFN2* p.Thr105Met mutant mice.

Another strategy targeting mitochondrial shape consists of the use of Szeto–Schiller (SS) peptides. SS peptides enter mitochondria and bind cardiolipin, a lipid of the inner mitochondrial membrane that regulates respiratory chain complexes and cristae structures. Elamipretide (also called MTP-131) is an SS peptide that has been found to be able to correct mitochondrial ultrastructure [202,203]. Again, this strategy has not yet been tested in MDDS but previous results suggest that it may constitute a potential approach for some forms of these syndromes.

A yeast-based screening assay of 1500 chemical compounds has identified clofilium tyosilate (CLO) to be useful to correct mtDNA instability in *POLG* mutants [204]. This compound was re-evaluated in human fibroblasts derived from one patient carrying *POLG* mutations, demonstrating its ability to rescue mtDNA levels [204]. Further studies with this compound have been conducted using zebrafish as a model, showing an increase in mtDNA levels either in mutant and wild-type individuals, as well as the restoration of complex I activity. Moreover, it improved the cardio-skeletal effects of *POLG* deficiency. Although it is known that CLO is an antiarrhythmic acting as a potassium channel blocker, its mechanism of action in rescuing mtDNA levels is not known [205].

## 5. Prospects and Specific Barriers

Many therapeutic approaches have been suggested or applied to treat mitochondrial diseases in general, and more specifically for MDDS. The complexity and variability of these disorders makes it difficult in many cases to definitely conclude whether they have demonstrated substantial efficacy. Those in clinical trial phases are expected to reach this milestone at the specific expected time points defined in the trial.

Currently, MDDS have very limited options for therapy, although significant progresses have been achieved in recent years and the situation is significantly different now than it was ten years ago. Some aggressive treatments, such as liver transplantation for mutations in *DGUOK*, *POLG*, *MPV17* and *TYMP*, or AHSCT for patients with mutations in *TYMP*, have shown variable success and to date are the only available choices for patients that, without treatment, could face a fatal outcome. One of the potential alternatives for some of these forms of MDDS is the implementation of gene therapy approaches. As indicated at the end of the corresponding Section 4.6, the investment needed for the mandatory preclinical and clinical investigations complying with the requirements of the regulatory agencies for gene therapy trials is an important barrier. However, it should be largely compensated by the potential benefits that these therapies will provide if they are demonstrated to be effective and safe for patients.

The potential improvement that may constitute the dN-treatment if it is finally approved for TK2 deficiency should be also underlined. Robust clinical results show that this therapy changes the clinical course of the disease, at least for paediatric patients [178]. Importantly, several preclinical studies at different in vitro and in vivo stages indicate that this approach may be also effective, for MDDS forms caused by mutations in *POLG*, *MPV17*, *DGUOK* or others [92,110,111,181,182,206]. However, the main barrier for this strategy is the poor availability of the orally administered dNs, especially the purines dAdo and dGuo. Further research to overcome this important difficulty should be conducted, including the inhibition of the fast catabolism of these compounds by several means, or exploring alternate administration routes that bypass or avoid the degradation of these molecules.

The special importance of the availability of in vivo models to test these therapies should be also noted. Experimental evidence supporting the efficacy of some of the experimental approaches discussed above has only been obtained in vitro, and we should be very cautious before using these approaches in patients, unless their efficacy is additionally confirmed in vivo using animal models. There are several genetically modified mouse strains that aim to model these diseases (reviewed by [207,208] as well as some other vertebrate models [206,209]). Some of the murine models recapitulate, at least partially, the clinical phenotype of the diseases [171,176,187,210,211,212], but in some other cases, the phenotype is barely biochemical or molecular, with no or little effect on the functional life of the animals [90,184]. Sometimes, the models can be stressed to show enhanced phenotypes that result as useful to show the efficacy of the experimental therapies [188,195,213]. In any case, testing in these models those treatments that have shown promising results in cell culture or other in vitro models should be encouraged in order to accelerate the implementation of clinical programs for them.

Lastly, there is a general barrier that contributes to complicating the advance of research and implementation of therapies for MDDS. All these are very rare disorders, with prevalences ranging from 1:51,000 for the Alpers–Huttenlocher syndrome [214] to ultrarare diseases such as MNGIE with around one case per 10 million people ([215] an unpublished personal estimate), or even less for other MDDS forms. This often makes it difficult to reach a correct diagnosis because of unawareness and hinders the recruitment of patients for the clinical investigation of new treatments. The rarity and severity of many MDDS also complicates the design of the clinical trials, and the clinical heterogeneity characteristic of MDDS also challenges the completion of informative clinical history studies, needed in some cases to substitute the absence of untreated control groups in clinical trials.

In recent years, research on rare diseases has gained increasing interest and resources, and this has flourished in a diversity of emerging therapies that could soon be available for patients. However, the clinical implementation of these therapies will ultimately depend on our ability to drive stakeholders’ attention toward the huge societal impact that these treatments would have. To strengthen and build up new wide and solid collaborative networks, including with basic and clinic researchers, will surely contribute to minimize many of the barriers challenging the development of future therapies for MDDS and other rare diseases.

## 6. Conclusions

The potential therapy choices for patients with mtDNA maintenance disorders have significantly increased in the last ten years. The data collected in this review indicate that there are currently nine active interventional clinical trials for seven different treatments, recruiting patients with mutations in five different genes (*POLG*, *TYMP*, *TK2*, *OPA1* and *SUCLA2*). This constitutes a good record if we consider that MDDS are very rare disorders. Moreover, there is growing optimism in the possibility of expanding the potential therapy options for these disorders due to the large number of preclinical data that researchers in the field have gathered in recent years. This is a growing group of diseases, with mutations in new genes causing MDDS found almost every year. However, they are well delimited by the presence of one or more of the hallmarks that define this group of disorders (i.e., somatic alterations of mtDNA such as depletion, multiple deletions and point mutations). Many of the genes associated with MDDS encode proteins belonging to the mtDNA replication machinery, dNTP metabolism or mitochondrial dynamics, and these biological processes are the subject of in-depth investigation by the scientific community working in the mitochondrial field. Without a doubt, the combination of relatively well-known pathomechanisms of these disorders and an increasing knowledge of the basic biochemical and molecular processes involved in mtDNA replication has contributed to the implementation of many novel therapy approaches for MDDS. Our efforts should now be dedicated to maintaining and further improving our knowledge in this field, thus generating new approaches and refining those already under preclinical and clinical investigation, and to overcome the aforementioned barriers hindering their final implementation into medical practice.

## Figures and Tables

**Figure 1 ijms-22-06447-f001:**
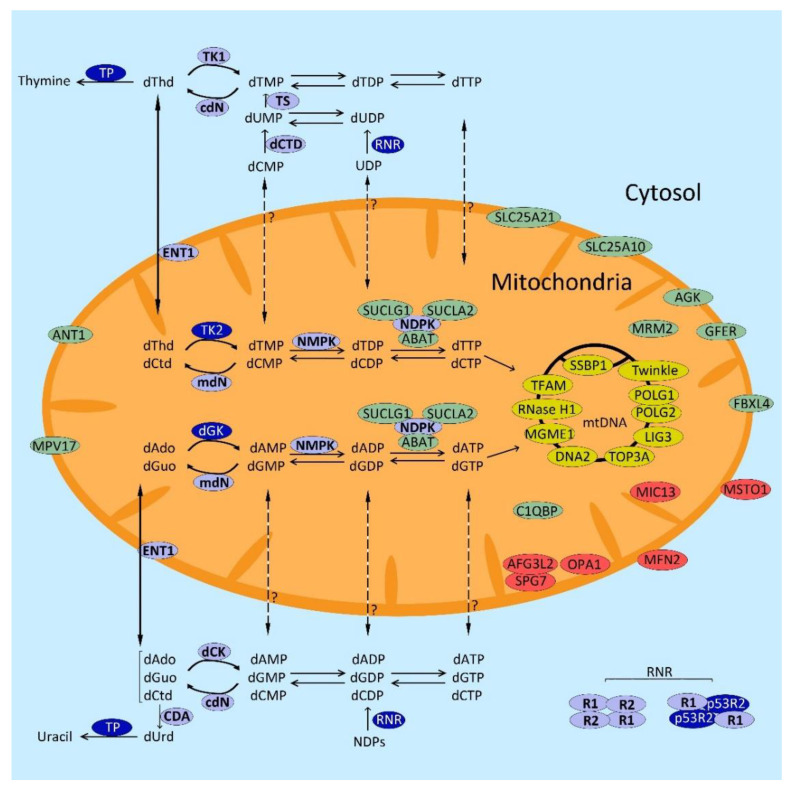
Metabolic pathways and proteins involved in mitochondrial DNA replication disorders. Proteins whose mutations have been linked to MDDS are depicted in yellow (mtDNA replication/maintenance machinery), dark blue (dNTP metabolism), red (mitochondrial dynamics) and green (unknown role in mtDNA replication). Light blue colour represents proteins participating in dNTP metabolism but not associated with MDDS. Abbreviations: ABAT, 4-aminobutyrate aminotransferase; AFG3L2, AFG3-like protein 2; AGK, acylglycerol kinase; ANT1, adenine nucleotide translocator 1; CDA, cytidine deaminase; cdN, cytosolic deoxyribonucleotidase; C1QBP, complement component 1 Q subcomponent-binding protein; dAdo, deoxyadenosine; dCK, deoxycytidine kinase; dCtd, deoxycytidine; dCTD, deoxycytidylate deaminase; dGK, deoxyguanosine kinase; dGuo, deoxyguanosine; DNA2, helicase/nuclease DNA2; dThd, thymidine; dUrd, deoxyuridine; ENT1, equilibrative nucleoside transporter 1; FBXL4, F-box/LRR-repeat protein 4; GFER, growth factor, augmenter of liver regeneration; LIG3, ligase III; mdN, mitochondrial deoxyribonucleotidase; MFN2, mitofusin 2; MGME1, mitochondrial genome maintenance exonuclease 1; MICOS13, MICOS complex subunit MIC13; MPV17, protein MPV17; MRM2, rRNA methyltransferase 2; MSTO1, protein misato homolog 1; NDPK, nucleoside diphosphate kinase; NMPK, nucleoside monophosphate kinase; OPA1, dynamin-like 120 kDa protein; POLG1, catalytic subunit of polymerase gamma; POLG2, ancillary subunit of polymerase gamma; p53R2; p53-inducible small subunit of the ribonucleotide reductase; R2, small subunit of the ribonucleotide reductase; RNASEH1, ribonuclease H1; RNR, ribonucleotide reductase; SLC25A10, mitochondrial dicarboxylate carrier; SLC25A21, mitochondrial 2-oxodicarboxylate carrier; SPG7, paraplegin; SSBP1, mitochondrial single strand binding protein; SUCLA2, β-subunit of the succinate-CoA ligase; SUCLG1, α-subunit of the succinate-CoA ligase; TFAM, mitochondrial transcription factor 1; TK1, thymidine kinase 1; TK2, thymidine kinase 2; TOP3A, DNA topoisomerase 3 alpha; TP, thymidine phosphorylase; TS, thymidylate synthase; Twinkle, mitochondrial helicase. Right bottom corner: the 2 isoforms of tetrameric RNR are represented, 2R1/2R2 (active in proliferating cells) and 2R1/2p53R2 (active throughout the cell cycle). Question marks indicate that the mitochondrial transporter(s) of the deoxyribonucleoside mono-, di- and triphosphates has or have not been identified. Figure generated using the Adobe^®^ Photoshop^®^ CS5 software.

**Table 1 ijms-22-06447-t001:** Genes associated with mitochondrial DNA replication disorders.

Category		Gene	Protein	Protein Function/Pathway	Clinical Features	Type of Inheritance	Type of mtDNA Aberration	OMIM # (Gene)	Reference * (Year)
**mtDNA replication machinery**		***POLG***	**DNA polymerase gamma**	**Polymerase**	Alpers–Huttenlocher syndrome/ataxia/PEO	AR/AD	D/MD/PM	174763	[16] (2001)
		***POLG2***	**DNA polymerase subunit gamma-2**	**Polymerase (ancillary)**	PEO/hepatic failure	AD/AR	MD/D	604983	[17] (2006)
		***TWNK***	**Twinkle**	**Helicase**	Perrault syndrome/PEO/ataxia/encephalopathy/IOSCA	AD/AR	D/MD/PM	606075	[18] (2001)
		***MGME1***	**Mitochondrial genome maintenance exonuclease 1**	**Exonuclease**	PEO/emaciation	AR	D/MD	615076	[19] (2013)
		***DNA2***	**DNA replication ATP-dependent helicase/nuclease DNA2**	**Helicase/nuclease**	PEO/myopathy/Seckel syndrome	AD	MD	601810	[20] (2013)
		***RNASEH1***	**Ribonuclease H1**	**Ribonuclease**	PEO/muscle weakness/dysphagia/spinocerebellar signs	AR	D/MD	604123	[21] (2015)
		***TFAM***	**Mitochondrial transcription factor A**	**Transcription factor**	Neonatal liver failure	AR	D	600438	[22] (2016)
		***TOP3A***	**DNA topoisomerase 3 alpha**	**Topoisomerase**	PEO/Bloom syndrome-like disorder	AR	MD/D	601243	[23] (2018)
		***SSBP1***	**Mitochondrial single strand binding protein**	**ssDNA stabilization**	Optic atrophy/liver failure/neurological syndrome /retinopathy	AD/AR	D	600439	[24] (2019)
		***LIG3***	**Ligase III**	**Mitochondrial DNA ligase**	MNGIE-like	AR	D	600940	[25] (2021)
**dNTP metabolism**		***TYMP***	**Thymidine phosphorylase**	**Nucleoside catabolism**	MNGIE	AR	D/MD/PM	603041	[26] (1999)
		***TK2***	**Thymidine kinase 2**	**dNTP anabolism**	Myopathy/PEO	AR	D/MD	188250	[27] (2001)
		***DGUOK***	**Deoxyguanosine kinase**	**dNTP anabolism**	Neurohepatopathy/myopathy/PEO	AR	D/MD	601465	[28] (2001)
		***RRM2B***	**p53-subunit of ribonucleotide reductase**	**dNTP anabolism**	Encephalomyopathy/PEO /MNGIE/KSS/neuropathy/deafness/tubulopathy	AR / AD	D/MD	604712	[29] (2007)
**Mitochondrial dynamics**		***OPA1***	**Dynamin-like 120 kDa protein, mitochondrial**	**GTPase/mitochondrial fusion**	Optic atrophy/Behr syndrome	AD	MD	605290	[30] (2008)
		***MFN2***	**Mitofusin-2**	**GTPase/mitochondrial fusion**	Optic atrophy/myopathy/axonal neuropathy/Charcot-Marie-Tooth	AR / AD	D/MD	608507	[31] (2012)
		***SPG7***	**Paraplegin**	**Subunit of m-AAA protease**	PEO/spastic paraplegia	AR	MD	602783	[32] (2014)
		***AFG3L2***	**AFG3-like protein 2**	**Subunit of m-AAA protease**	PEO/ataxia	AD	MD	604581	[33] (2015)
		***MSTO1***	**Protein misato homolog 1**	**Mitochondrial fusion**	Muscular dystrophy with cerebellar involvement/myopathy/ataxia	AR	D	617619	[34] (2017)
		***MICOS13***	**MICOS complex subunit MIC13**	**Maintenance of cristae structure**	Hepato-encephalopathy	AR	D	616658	[35] (2019)
**Unknown pathomechanism**	**Membrane channels**	***SLC25A4***	**Adenine nucleotide translocator**	**ADP/ATP carrier**	PEO/cardiomyopathy/myopathy	AD / AR	MD	103220	[36] (2000)
		***MPV17***	**Protein mpv17**	**Membrane channel/unknown**	Neurohepatopathy/neuropathy/leukoencephalopathy/Charcot–Marie–Tooth	AR	D/MD	137960	[37] (2006)
		***SLC25A21***	**Mitochondrial 2-oxodicarboxylate carrier**	**Transmembrane transporter**	Spinal muscular atrophy-like	AR	D	607571	[38] (2018)
		***SLC25A10***	**Mitochondrial dicarboxylate carrier**	**Transmembrane transporter**	Epileptic encephalopathy	AR	D	606794	[39] (2018)
	**Other function / unknown function**	***SUCLA2***	**β-subunit, Succinate-CoA ligase**	**Krebs cycle**	Encephalomyopathy	AR	D	603921	[40] (2005)
		***SUCLG1***	**α-subunit, Succinate-CoA ligase**	**Krebs cycle**	Encephalomyopathy	AR	D	611224	[41] (2007)
		***AGK***	**Acylglycerol kinase**	**Lipid metabolism**	Congenital cataract/hypertrophic cardiomyopathy/skeletal myopathy and lactic acidosis/Sengers syndrome	AD	D	610345	[42] (2012)
		***GFER***	**Growth factor, augmenter of liver regeneration**	**Growth factor**	Progressive myopathy/congenital cataract/sensorineural hearing loss/developmental delay	AR	MD	600924	[43] (2009)
		***ABAT***	**4-aminobutyrate aminotransferase**	**Aminotransferase**	Encephalomyopathy	AR	D	137150	[44] (2015)
		***FBXL4***	**F-box/LRR-repeat protein 4**	**Protein homeostasis**	Encephalomyopathy	AR	D	605654	[45] (2013)
		***MRM2***	**rRNA methyltransferase 2, mitochondrial**	**Mito rRNA maturation**	MELAS-like	AR	D	606906	[46] (2017)
		***C1QBP***	**Complement component 1 Q subcomponent-binding protein, mitochondrial**	**Inflammation/nuclear transcription/mitoribosome biogenesis/apoptosis**	Cardiopathy-multisystemic/PEO-myopathy	AR	MD	601269	[47] (2017)

Abbreviations: AD, autosomal dominant; AR, autosomal recessive; D, mtDNA depletion; dNTP, deoxyribonucleoside triphosphate; IOSCA, infantile onset spinocerebellar ataxia; KSS, Kearns–Sayre syndrome; MD, mtDNA multiple deletions; MELAS, mitochondrial encephalomyopathy, lactic acidosis, and stroke-like episodes; MNGIE, mitochondrial neurogastrointestinal encephalomyopathy; PEO, progressive external ophthalmoplegia; PM, mtDNA somatic point mutations; ssDNA, single-stranded DNA. # Number assigned at the gene in the Online Mendelian Inheritance in Man database (https://www.omim.org/, accessed on 22 May 2021); * The first report associating mutations in the gene to a disease with mtDNA alterations.

**Table 2 ijms-22-06447-t002:** Clinical trials recruiting patients with mitochondrial DNA replication disorders. Only ongoing or recently conducted clinical trials accepting patients with MDDS are listed.

**NCT04378075**	**A Study to Evaluate Efficacy and Safety of Vatiquinone for Treating Mitochondrial Disease in Participants with Refractory Epilepsy**
Condition	*POLG*
Study type/phase	Interventional (phase 2 and phase 3)
Intervention	Vatiquinone administration
Status	Recruiting
Estimated study completion	1 April 2023
Outcomes	Change in the number of observable motor seizures Occurrence or recurrence of epilepsy Participants who require rescue seizure medication
Sponsor	PTC therapeutics
**NCT01370447**	**EPI-743 for Mitochondrial Respiratory Chain Diseases**
Condition	*POLG*
Study type/phase	Interventional (phase 2)
Intervention	EPI-743
Status	Active, not recruiting
Estimated study completion	31 December 2021
Outcomes	Change in neuromuscular function Number of subjects experiencing adverse events Change in Newcastle Paediatric Mitochondrial Disease Score Pharmacokinetics of EPI-743
Sponsor	PTC Therapeutics
**NCT02023866**	**Open-Label, Dose-Escalating Study Assessing Safety, Tolerability, Efficacy, of RP103 in Mitochondrial Disease**
Condition	*POLG-TYMP*
Study type/phase	Interventional (phase 2)
Intervention	Cysteamine bitartrate
Status	Completed
Estimated study completion	October 2016
Outcomes	Change in Newcastle Paediatric Mitochondrial Disease Scale (NPMDS) Score
Sponsor	Horizon Pharma USA, Inc.
**NCT02473445**	**A Long-Term Extension of Study RP103-MITO-001 (NCT02023866) to Assess Cysteamine Bitartrate Delayed-Release Capsules (RP103) in Children with Inherited Mitochondrial Disease**
Condition	*POLG-TYMP*
Study type/phase	Interventional (phase 2)
Intervention	Cysteamine bitartrate
Status	Completed
Estimated study completion	6 March 2017
Outcomes	Change in Newcastle Paediatric Mitochondrial Disease Scale (NPMDS) Score
Sponsor	Horizon Pharma USA, Inc.
**NCT03701568**	**A RETROspective Study of Patients with TK2d**
Condition	*TK2*
Study type/phase	Observational
Intervention	dCtd/dThd
Status	Completed
Estimated study completion	31 May 2019
Outcomes	Clinical course Motor function and ambulatory assessments
Sponsor	Modis Therapeutics, Inc.
**NCT03845712**	**An Open-Label Study of Continuation Treatment with Combination Pyrimidine Nucleosides in Patients With TK2**
Condition	*TK2*
Study type/phase	Interventional (phase 2)
Intervention	MT1621
Status	Active, not recruiting
Estimated study completion	31 January 2022
Outcomes	Safety Motor function assessments Respiratory status Growth/nutrition Pharmacokinetics Quality of life through patient questionnaire
Sponsor	Modis Therapeutics, Inc.
**NCT04581733**	**A Study of the Efficacy and Safety of MT1621 in Thymidine Kinase 2 (TK2) Deficiency**
Condition	*TK2*
Study type/phase	Interventional (phase 3)
Intervention	MT1621
Status	Not yet recruiting
Estimated study completion	March 2025
Outcomes	Time to loss/acquisition of any motor milestone Overall survival
Sponsor	Modis Therapeutics, Inc.
**NCT03639701**	**Treatment of TK2 Deficiency with Thymidine and Deoxycytidine**
Condition	*TK2*
Study type/phase	Interventional (phase 1 and phase 2)
Intervention	dThd
Status	Enrolling by invitation
Estimated study completion	1 April 2024
Outcomes	Safety Efficacy measured by different ways
Sponsor	Columbia University
**NCT03866954**	**Trial of Erythrocyte Encapsulated Thymidine Phosphorylase in Mitochondrial Neurogastrointestinal Encephalomyopathy**
Condition	*TYMP*
Study type/phase	Interventional (phase 2)
Intervention	EETP
Status	Not yet recruiting
Estimated study completion	September 2022
Outcomes	Safety of procedure Pharmacodynamic effects Efficacy of EETP Changes in clinical assessments
Sponsor	St George’s, University of London
**NCT02427178**	**MNGIE Allogeneic Hematopoietic Stem Cell Transplant Safety Study**
Condition	*TYMP*
Study type/phase	Interventional (phase 1)
Intervention	Hematopoietic allogenic stem cells
Status	Recruiting
Estimated study completion	June 2023
Outcomes	Engraftment success Survival Blood levels of dThd and dUrd
Sponsor	Columbia University
**NCT00804102**	**Transcorneal Electrical Stimulation Therapy for Retinal Disease**
Condition	*OPA1*
Study type/phase	Interventional (not phase applicable)
Intervention	Transcorneal electrical stimulation
Status	Completed
Estimated study completion	April 2011
Outcomes	Enhanced field of vision Enhanced visual acuity Lower threshold for electrical evoked phosphenes
Sponsor	Okuvision GmbH
**NCT03011541**	**Stem Cell Ophthalmology Treatment Study II**
Condition	*OPA1*
Study type/phase	Interventional (not phase applicable)
Intervention	Administration of autologous bone marrow derived stem cells
Status	Recruiting
Estimated study completion	January 2022
Outcomes	Visual acuity Visual fields Optical coherence tomography
Sponsor	MD Stem Cells
**NCT01648634**	**Nebivolol for the Prevention of Left Ventricular Systolic Dysfunction in Patients with Duchenne Muscular Dystrophy**
Condition	*SUCLA2*
Study type/phase	Interventional (phase 3)
Intervention	Nevibolol
Status	Active, not recruiting
Estimated study completion	June 2021
Outcomes	Left ventricular systolic dysfunction Right ventricular ejection fraction Hospitalizations Mortality
Sponsor	Assistance Publique-Hôpitaux de Paris

Abbreviations: dCtd, deoxycytidine; dUrd, deoxyuridine; dThd, thymidine; EETP, erythrocyte-entrapped thymidine phosphorylase; EPI-743, vatiquinone, alpha-tocotrienol quinone; MNGIE, mitochondrial neurogastrointestinal encephalomyopathy; MT1621, combination of thymidine and deoxycytidine for oral administration; *OPA1*, gene encoding the dynamin-like 120 kDa protein, mitochondrial; *POLG*, gene encoding the catalytic subunit of polymerase gamma; *TK2*, gene encoding thymidine kinase 2; *TYMP*, gene encoding thymidine phosphorylase; *SUCLA2*, gene encoding the β-subunit of succinate-CoA ligase. Data obtained from https://clinicaltrials.gov/ (accessed on 22 May 2021).

## Data Availability

Not applicable.

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
