# Peer review of "Therapy Prospects for Mitochondrial DNA Maintenance Disorders"

_ijms, 2021, doi:10.3390/ijms22126447_

Round 1

Reviewer 1 Report

This manuscript is a review of therapies for disorders related to mitochondrial DNA management. The two such disorders are Mitochondrial DNA depletion and multiple deletions syndromes (MDDS). The manuscript covers recent literature (> 70% of cited articles) on therapies for these diseases. The therapies in clinical stage are also discussed.

The manuscript is well organized and presented. I believe the topic is emerging and will attract wide readership.

Table 2: Please mention the source of the clinical data and accessed date.

Author Response

We thank the reviewer for his/her time in reading the manuscript and making us aware of aspects to be improved. We have followed your recommendations to generate the revised version of the manuscript.

This manuscript is a review of therapies for disorders related to mitochondrial DNA management. The two such disorders are Mitochondrial DNA depletion and multiple deletions syndromes (MDDS). The manuscript covers recent literature (> 70% of cited articles) on therapies for these diseases. The therapies in clinical stage are also discussed.

The manuscript is well organized and presented. I believe the topic is emerging and will attract wide readership.

Table 2: Please mention the source of the clinical data and accessed date.

We have added a caption at the bottom of Table 2 in the revised version of the manuscript, where this information is quoted.

Reviewer 2 Report

The bibliographic review carried out by the authors is very interesting, the writing and reading is pleasant, allowing a good understanding of the text. The topic of the review is current and its deepening is essential to address such important aspects of human health. One aspect to correct would be the incorrect affiliation of (1), which does not indicate that the country to which Barcelona belongs is Spain. The authors indicate Catalonia as a country, and that is not true.

Some aspects could be improved.

1-section 1. Numerous bibliographic citations to the numerous statements made by the authors are missing.

2- p2. Regarding the origin of the genetic material of the mitochondria, perhaps the authors are too forceful in their statements.

  1. The meaning of # is missing from table 1.
  2. Fig. 1. “caption” should be removed. The program used to create the figure should be indicated.
  3. Table 2. Abbreviations are missing at the base of the table.
  4. section 4.1. The meaning of MNGIE should be clarified.
  5. sectoin 4.4. Line 486. Authors should cite correctly.
  6. A definite conclusion of the review is necessary.

Author Response

We thank the reviewer for his/her time in reading the manuscript and making us aware of aspects to be improved. We have followed your recommendations to generate the revised version of the manuscript.

Some aspects could be improved.

1-section 1. Numerous bibliographic citations to the numerous statements made by the authors are missing.

We agree with the reviewer. We have included bibliographic references for our statements in section 1. Because many of these statements are general or detailed information derived from knowledge gained in a myriad of different primary works (e.g., the number of genes contained in human mtDNA and what they encode) many of the references of this section are previous reviews from authorized experts compiling this information.

2-p2. Regarding the origin of the genetic material of the mitochondria, perhaps the authors are too forceful in their statements.

We agree with the reviewer. We have smoothed the statement about the origin of the mitochondrial genome and added a couple of references documenting our modified statement.

3-The meaning of # is missing from table 1.

This meaning has been included at the caption of Table 1

4-Fig. 1. “caption” should be removed. The program used to create the figure should be indicated.

These recommendations have been followed.

5-Table 2. Abbreviations are missing at the base of the table.

A caption has been included at the bottom of table 1 where the abbreviations are expanded.

6-section 4.1. The meaning of MNGIE should be clarified.

The meaning of MNGIE is expanded at this point.

7-sectoin 4.4. Line 486. Authors should cite correctly.

The document referred at this point has been cited. However, since this is not a physical document nor a PDF, but an internet document periodically updated, we have maintained the link to the website and specified the last update at the time of the generation of this manuscript.

8-A definite conclusion of the review is necessary.

A final section with the conclusions has been added to the manuscript.
